

# Characterization of organic nitrogen in aerosols at a forest site in the southern Appalachian Mountains

Xi Chen[1], Mingjie Xie[2,a], Michael D. Hays[1], Eric Edgerton[3], Donna Schwede[4], John T. Walker[1,*]

[1]National Risk Management Research Laboratory, Office of Research and Development, U.S. Environmental Protection Agency, Research Triangle Park, North Carolina, 27711, U.S.A.

[2]Oak Ridge Institute for Science and Education (ORISE), National Risk Management Research Laboratory, Office of Research and Development, U.S. Environmental Protection Agency, Research Triangle Park, North Carolina, 27711, U.S.A.

[3]Atmospheric Research and Analysis, Inc., Cary, NC, 27513

[4]National Exposure Research Laboratory, Office of Research and Development, U.S. Environmental Protection Agency, Research Triangle Park, North Carolina, 27711, U.S.A.

[a]Present address: School of Environmental Science and Engineering, Nanjing University of Information Science & Technology, Nanjing 210044, China

[*]Corresponding Author: Tel.: +1 919 541 2288.  Email:Walker.JohnT@epa.gov.

**Abstract**

This study investigates the composition of organic particulate matter in a remote montane forest in the southeastern U.S., focusing on the role of organic nitrogen (N) in sulfur-containing secondary organic aerosol (nitrooxy-organosulfates) and aerosols associated with biomass burning (nitro-aromatics).  Bulk water soluble organic N (WSON) represented ~ 14% w/w of water soluble total N (WSTN) in $PM_{2.5}$, on average, across seasonal measurement campaigns conducted in the spring, summer, and fall of 2015. Largest contributions of WSON to WSTN were observed in spring (~ 18% w/w) and lowest in the fall (~10% w/w). On average, identified nitro-aromatic and nitrooxy-organosulfate compounds accounted for a small fraction of WSON, ranging from ~ 1% in spring to ~ 4% in fall, though were observed to contribute as much as 28% w/w of WSON in individual samples. Highest concentrations of oxidized organic N species occurred during summer (average of $0.65 ngN/m^3$) along with a greater relative abundance of higher generation oxygenated terpenoic acids, indicating an association with more aged aerosol.



Highest concentrations of nitro-aromatics (eg. nitrocatechol and methyl-nitrocatechol),
levoglucosan, and aged SOA tracers were observed during fall, associated with aged biomass
burning plumes.  Nighttime nitrate radical chemistry is the most likely formation pathway for
nitrooxy-organosulfates observed at this low NOx site (generally <1ppb). Isoprene derived
organosulfate (MW216, 2-methyltetrol derived), which is formed from isoprene epoxydiols
(IEPOX) under low NOx conditions, was the most abundant individual organosulfate.
Concentration weighted average N/C ratios for nitro-aromatics + organosulfates + terpenoic
acids were one order of magnitude lower than the overall aerosol N/C ratio, indicating the
presence of other uncharacterized higher N content species. Although nitrooxy-organosulfates
and nitro-aromatics contributed a small fraction of WSON, our results provide new insight into
the atmospheric formation processes and sources of these largely uncharacterized components of
atmospheric organic N, which also helps to advance the atmospheric models to better understand
the chemistry and deposition of reactive N.

**1.  Introduction**

There is extensive evidence showing that boreal and temperate forests are affected by

anthropogenic activities, both industrial and agricultural. Such activity results in unprecedented
quantities of reactive nitrogen (N) being released into the atmosphere, subsequently altering
global nitrogen and carbon (C) biogeochemical cycles (Bragazza et al., 2006; Doney et al., 2007;
Ollinger et al., 2002; Magnani et al., 2007; Neff et al., 2002a,b; Pregitzer et al., 2008). Nitrogen
enters natural ecosystems through atmospheric deposition and biological fixation, and is mainly
lost through leaching and gaseous fluxes back to the atmosphere (Hungate et al., 2003).
Atmospheric deposition of N to terrestrial ecosystems may lead to soil and aquatic acidification,
nutrient imbalance and enrichment, plant damage and microbial community changes as well as
loss of biodiversity (Bobbink et al., 1998; Magnani et al., 2007; Lohse et al, 2008; Simkin et al.,

2016).

In the United States, deposition of atmospheric pollutants including N is monitored by

the National Atmospheric Deposition Program (NADP) and EPA's Clean Air Status and Trends
Network (CASNET). However, these networks focus only on inorganic N species (eg.
$NH_3/NH_4^+$ and $HNO_3/NO_3^-$). Recent studies shed light on the importance of organic N
deposition, which is not routinely measured in national networks.  On a global basis, organic N



may contribute ~ 25 percent of the total N deposition (Gonzalez Benitez et al., 2009; Jickells et
al., 2013; Keene et al., 2002; Neff et al., 2002a; Zhang et al., 2012). Although ubiquitous,
widespread routine monitoring of organic N in the atmosphere is inhibited due to difficulties in
sampling (Walker et al, 2012) and inability to fully speciate the wide range of constituents that
make up this large pool of atmospheric N (Altieri et al., 2009; Cape et al., 2011; Neff et al.,
2002a; Samy et al., 2013). For these reasons, understanding of the sources, atmospheric
chemistry, and deposition of organic nitrogen remains limited.

Atmospheric N from biogenic and anthropogenic emissions sources undergoes complex

transformation processes and photochemical reactions. Consequently, apportionment of
atmospheric organic N to potential sources is challenging. However, such information is required
to advance atmospheric N models applied to better understand the global N cycle.  For example,
Miyazaki et al. (2014) examined aerosols collected in a deciduous forest and found in the
summer that water soluble organic N (WSON) correlated positively to biogenic hydrocarbon
oxidation; and during fall WSON in the coarse particle fraction was associated with primary
biological emissions (e.g. emitted from soil, vegetation, pollen and bacteria). Such patterns
underscore that atmospheric organic N measured in forested landscapes originates from a variety
of sources that contribute differently across seasons.

Recent advancements have been made in speciation of organic N in aerosol for some

groups of compounds including amines, amino acids and other nitrogenated functional groups
such as organonitrates (Day et al., 2010; Place et al., 2017; Samy et al., 2013). Organic N in
secondary aerosol and aerosols associated with biomass burning sources are areas of increasing
interest, from both atmospheric chemistry and ecosystem exposure perspectives, where more
information is needed. Studies of secondary organic aerosols (SOA) have identified a variety of
nitrated organosulfate compounds (e.g. nitrooxy-organosulfates) in both chamber and ambient
aerosol samples following isoprene and monoterpenes oxidation. These compounds are either
produced under high NOx conditions or from nighttime $NO_3$ radical chemistry (Surratt et al.,
2006, 2007, 2008, 2010; Darer et al., 2011; Lin et al., 2013a; He et al., 2014; Worton et al.,
2013).  Potential SOA precursors such as unsaturated green leaf volatiles (GLVs) released by
wounded plants (e.g. crop harvesting and insect attacks) may contribute substantially to the
budget of biogenic SOA formation especially in remote forests (Gomez-Gonzalez et al., 2008;
Hamilton et al., 2013; Shalamzari et al., 2016).  The detection of reaction products such as



organosulfates and nitrooxy-organosulfates in ambient aerosols provides strong evidence of
influence from anthropogenic sources (e.g. $SO_2$ and $NO_x$) interacting with biogenic precursors to
form nitrogenated SOA (Chan et al., 2010; Lin et al., 2013a; Meade et al., 2016).
In addition to being present in sulfur-containing SOA, organic nitrogen, specifically
nitro-aromatic compounds (e.g.nitrophenols and nitrocatechols ), have been characterized as
chemical tracers from biomass burning (e.g. wildland and prescribed smoke, bushfires,
residential wood burning). This is in addition to levoglucosan, a widely used tracer of biomass
burning (Iinuma et al., 2010, 2016; Kahnt et al., 2013; Kitanovski et al., 2012; Gaston et al.,
2016). These nitrated compounds can form during pyrolysis of plant biopolymers such as
cellulose. Furthermore, as combustion byproducts, these compounds are often defined as brown
carbon (BrC) and thus potentially light absorbing (Mohr et al., 2013; Liu et al., 2015).
Presumably, nitro-aromatics could constitute a substantial portion of atmospheric organic N in
aerosols collected in regions affected by biomass burning.
This study investigates the composition of organic particulate matter in a remote montane
forest in the southeastern U.S., focusing on the role of organic N in sulfur-containing SOA and
aerosols associated with biomass burning. Measurements target four groups of compounds: 1)
nitro-aromatics associated with biomass burning; 2) organosulfates and nitrooxy-organosulfates
produced from biogenic SOA precursors (i.e., isoprene, monoterpenes and unsaturated
aldehydes) interacting with anthropogenic pollutants; 3) terpenoic acids formed from
monoterpene oxidation; and 4) organic molecular markers including methyltetrols, C-5 alkene
triols, 2-methylglyceric acid, 3-hydroxyglutaric acid and levoglucosan. Terpenoic acids and
organic markers are included to assist in characterizing the extent of biogenic compound
oxidation and atmospheric processing (i.e., aerosol aging) as well as contributions from biomass
burning sources. Aerosol bulk chemical measurements are conducted to estimate total water
soluble organic N and C concentrations. Characterization of seasonal patterns in concentrations
of organic N species and assessment of potential sources and formation processes are
emphasized.

**2. Experimental methods and materials**
2.1 Sampling site and atmospheric aerosol collection



129  The study was conducted at the U.S. Forest Service Coweeta Hydrologic Laboratory, a

130 2185-ha experimental forest in southwestern, North Carolina, USA (35°3' N, 83°25' W) near the

131 southern end of the Appalachian Mountain chain. The climate is classified as maritime, humid

132 temperate, with mean monthly temperatures ranging from 3.3°C in January to 21.6°C in July

133 (Swift et al., 1988).  Elevation ranges from 675 to 1592 m with a corresponding range in annual

134 precipitation of 1800 to 2500 mm (Swank and Crossley, 1988). The vegetation is characterized

135 as mixed coniferous/deciduous including oak, pines, and hardwoods (Bolstad et al, 1998).

136 Atmospheric measurements were conducted in the lowest part of the basin (686 m), collocated

137 with long term measurements of air and precipitation chemistry conducted by CASTNET and

138 NADP networks, respectively.

139  The sampling site is 5 km west of Otto, NC (population 2500) and Highway 23 (Figure

140 S1, supplemental material).  Land to the north, west and south of Coweeta is undeveloped forest.

141 Typical rural development is present to the east of the site, consisting of houses and small scale

142 farming for hay and crop production including some scattered cow and horse pastures.  The

143 nearest metropolitan areas include Atlanta, Georgia (175 km southwest), Chattanooga,

144 Tennessee (175 km west), Knoxville, Tennessee (110 km north/northwest), Asheville, North

145 Carolina (100 km northeast), and Greeneville, South Carolina (100 km southeast). The location

146 of the sampling site within the context of NOx and $SO_2$ point sources in the eastern U.S. is

147 shown in supplemental material (Figure S2).  Only minor point sources are present within ~ 100

148 km of the site.

149  The study period summarized here comprises three seasonal intensives conducted during

150 the spring, summer and fall of 2015. Each campaign was conducted for approximately 3 weeks

151 (21 May to 9 June, 6 August to 25 August, 9 October to 26 October). A high-volume Tisch TE-

152 1000 (Tisch Environmental, Cleves, OH) dual cyclone $PM_{2.5}$ sampler operated at a flow rate of

153 230 L/min was set up on the ground to collect 24 hr (started at 7am local time) integrated

154 samples on pre-baked (550°C for 12hrs) quartz fiber (QF) filters (90mm, Pall Corporation, Port

155 Washington, NY). Field blanks were collected the same way except being loaded in the sampler

156 without the pump switched on.  A total of 58 ambient samples and 10 field blanks were obtained.

157 Collected filter samples were transferred back to the laboratory in a cooler and stored in a freezer

158 at -20 °C before chemical analysis.

159  2.2 Trace gas and meteorological measurements



During the spring 2015 campaign, NOx concentrations were measured on a short tower
(7 m above ground) co-located with the CASTNET and high volume PM samplers. NOx
concentrations were measured using a commercial NO-NO2-NOx analyzer (model 42S, Thermo
Environmental Instruments, Incorporated, Franklin, MA). Briefly, nitric oxide (NO) is measured
directly on one channel by chemiluminescence. On a 2$^{nd}$ channel, NO2 is converted to NO by a
molybdenum catalyst heated to 325ºC, yielding the concentration of NOx (NO + NO2).  This
approach may overestimate NOx since other oxidized nitrogen gases such as HNO3, PAN and
HONO could also be reduced to NO on the heated molybdenum surface (Fehsenfeld et al., 1987;
Williams et al.,1998; Zellweger et al., 2000). However, the use of an inlet filter and
approximately 12 m of sample line between the atmospheric inlet and converter likely minimized
the potential bias from HNO3.  For subsequent campaigns, NOx concentrations were estimated
from a co-located NOy analyzer. Similar to the NOx instrument, NOy and HNO3 were also
measured using a modified model 42S NO-NO2-NOx analyzer. The NOy technique is described
in detail by Williams et al. (1998). Briefly, total oxidized reactive nitrogen (NOy) is converted to
NO using a molybdenum catalyst heated to 325ºC. On a 2$^{nd}$ channel, a metal denuder coated with
potassium chloride (KCl) is used to remove HNO3 before passing through a 2$^{nd}$ molybdenum
converter heated to 325ºC.  The difference between the total NOy measurement and the HNO3-
scrubbed NOy measurement is interpreted as HNO3. NOx concentrations were estimated from
the differences between measured NOy and HNO3, which provided an upper bound estimation as
gaseous N containing species were not excluded (eg. PAN and organic nitrates). Hourly ozone
concentrations were measured by CASTNET (U.S. EPA, 2017) on a co-located 10m tower.
Hourly meteorological data were provided by CASTNET (U.S. EPA, 2017) and Forest Service
(Miniat et al 2015; Oishi et al.,2017), including temperature, relative humidity, solar radiation
and precipitation.

2.3 Chemical analysis
2.3.1 Elemental and organic carbon analysis
A 1.5cm$^2$ QF punch was analyzed for elemental carbon (EC) and organic carbon (OC) using
a thermo-optical transmittance (TOT) method (Sunset Laboratory Inc, Oregon, USA) (Birch and
Cary, 1996).



2.3.2 Water soluble species by Ion Chromatography (IC) and Total Organic Carbon/Total
Nitrogen (TOC/TN) analyzers
A second QF punch ($1.5 cm^2$) from each sample was extracted with DI water (18.2
MΩ·cm, Milli-Q Reference system, Millipore, Burlington, MA) in an ultrasonic bath for 45 min.
The sample extract was filtered through a 0.2μm pore size PTFE membrane syringe filter (Iso-
disc, Sigma Aldrich, St. Louis, MO) before subsequent analyses.
Water soluble organic carbon (WSOC) and total N (WSTN) concentrations were
measured using a chemiluminescence method that included a total organic carbon analyzer
(TOC-Vcsh) combined with a total nitrogen module (TNM-1) (Shimadzu Scientific Instruments,
Columbia, MD).  For WSOC measurements, 25% phosphoric acid was mixed with sample
extract (resulting in a 1.5% acid mixture) and sparged for 3 min to remove any existing
carbonate/bicarbonate.
Inorganic species ($NH_4^+$, $NO_3^-$, $NO_2^-$ and $SO_4^{2-}$) were analyzed using ion chromatography
(IC, Dionex model ICS-2100, Thermo Scientific, Waltham, MA). The IC was equipped with
guard (IonPac 2mm AG23) and analytical columns (AS23) for anions. The samples were
analyzed using an isocratic eluent mix carbonate/bicarbonate (4.5/0.8mM) at a flow rate of 0.25
mL/min.  Cations were analyzed by Dionex IonPac 2mm CG12 guard and CS12 analytical
column**s**; separations were conducted using 20mM methanesulfonic acid (MSA) as eluent at a
flow rate of 0.25mL/min.  Multi-point (≥5) calibration was conducted using a mixture prepared
from individual inorganic standards (Inorganic Ventures, Christiansburg, VA). A mid-level
accuracy check standard was prepared from certified standards mix (AccuStandard, New Haven,
CT) for quality assurance/quality control purposes.

2.3.3 UV-Vis light absorption analysis
Several studies have shown that methanol can extract aerosol OC at higher efficiencies
than water, and that a large fraction of light absorption in the near-UV and visible ranges is
ascribed to water insoluble OC (Chen and Bond, 2010; Liu et al., 2013; Cheng et al., 2016). In
this study, a QF punch ($1.5 cm^2$) was extracted with 5 mL methanol (HPLC grade, Thermo
Fisher Scientific Inc.) in a tightly closed amber vial, sonicated for 15 min, and then filtered
through a 0.2 μm pore size PTFE filter (Iso-disc, Sigma Aldrich, St. Louis, MO). The light
absorption of filtered extracts was measured with a UV-Vis spectrometer over λ = 200-900 nm at



0.2 nm resolution (V660, Jasco Incorporated, Easton MD). The wavelength accuracy is less than
± 0.3 nm; the wavelength repeatability is less than ± 0.05 nm. A reference cuvette containing
methanol was used to account for solvent absorption. The UV-Vis absorption of field blank
samples was negligible compared to ambient samples, but used for correction nonetheless.  For
ease of analysis, the absorption at 365 nm referencing to absorption at 700 nm was used as a
general measure of the absorption by all aerosol chromophore components (Hecobian et al.,

2010).


2.3.4. Analysis of isoprene and monoterpene SOA markers and anhydrosugars by GC-MS
Aliquots of each filter (roughly ¼) were extracted by 10 mL of methanol and methylene
chloride mixture (1:1, v/v) ultrasonically twice (15 minutes each). The total extract was filtered
and concentrated to a final volume of ~0.5 mL. Next, extracts were transferred to a 2 mL glass
vial and concentrated to dryness under a gentle stream of ultrapure $N_2$ and reacted with 50 μL of
N, O-bis(trimethylsilyl)trifluoroacetamide (BSTFA) containing 1% trimethylchlorosilane
(TMCS) and 10 μL of pyridine for 3 h at 70 °C. After cooling down to room temperature,
internal standards (mixture of 17.6 ng μL$^{-1}$ acenaphthalene-d10 and 18.6 ng μL$^{-1}$ pyrene-d4
mixed in hexane) and pure hexane were added. The resulting solution was analyzed by an
Agilent 6890N gas chromatograph (GC) coupled with an Agilent 5975 mass spectrometer (MS)
operated in the electron ionization mode (70 eV). An aliquot of 2 μL of each sample was injected
in splitless mode. The GC separation was carried out on a DB-5 ms capillary column (30 m ×
0.25 mm × 0.25 μm, Agilent Technologies, Santa Clara, CA). The GC oven temperature was
programmed from 50 °C (hold for 2 min) to 120 °C at 30 °C min$^{-1}$ then ramped at 6 °C min$^{-1}$ to
a final temperature of 300 °C (hold for 10 min). Linear calibration curves were derived from six
dilutions of quantification standards. Anhydrosugars (levoglucosan) were quantified using
authentic standard; 2-methyltetrols (2-methylthreitol and 2-methylerythritol) and C-5 alkene
triols were quantified using meso-erythritol; other SOA tracers (e.g., hydroxyl dicarboxylic acid)
were quantified using cis-ketopinic acid (KPA) (refer to supplemental information Table S1).
The species not quantified using authentic standards were identified by the comparison of mass
spectra to previously reported data (Claeys, et al., 2004, 2007; Surratt et al., 2006; Kleindienst
al., 2007). Field blanks were collected and no contamination was observed for identified species.



2.3.5. Analysis of organosulfates, terpenoic acids and nitro-aromatics by High Performance
Liquid Chromatography- electrospray ionization-Quadrupole time-of-flight-Mass
Spectrometer (HPLC-ESI(-)-QTOF-MS)
Approximately 3-5 mL of methanol was used to ultrasonically extract (twice for 15 min)
roughly half of each 90mm QF sample. Internal standards (I.S.) were spiked onto each filter
sample prior to extraction (refer to Table S2, S3 and S4 for individual compounds and surrogate
standards used for each group of compounds). Extracts were filtered into a pear-shaped glass
flask (50 mL) and rotary evaporated to ~0.1 mL.  The concentrated extracts were then transferred
into a 2 mL amber vial that was rinsed with methanol 2-3 times. The final sample extract volume
was ~500 µL prior to analysis. All the glassware used during the extraction procedure was pre-
baked at 550°C overnight. Extracted samples were stored at or below -20 °C prior to analysis and
typically analyzed within 7 d.
An HPLC coupled with a quadrupole time-of-flight mass spectrometer (1200 series LC
and QTOF-MS, Model 6520, Agilent Technologies, Palo Alto, CA) was used for target
compound identification and quantification. The QTOF-MS instrument was equipped with a
multimode ion source operated in electrospray ionization (ESI) negative (-) mode. Optimal
conditions were achieved under parameters of 2000 V capillary voltage, 140 V fragmentor
voltage, 65 V skimmer voltage, 300 °C gas temperature, 5 L/min drying gas flow rate and 40
psig nebulizer. The ESI-QTOF-MS was operated over the m/z range of 40 to 1000 at a 3
spectra/s acquisition rate. Target compounds separation was achieved by a C18 column (2.1×100
mm, 1.8 µm particle size, Zorbax Eclipse Plus, Agilent Technologies) with an injection volume
of 2 µL and flow rate of 0.2 mL/min. The column temperature was kept at 40 °C, and gradient
separation was conducted with 0.2% acetic acid (v:v) in water (eluent A) and methanol (eluent
B). The eluent B was maintained at 25% for the first 3 min, increased to 100% in 10 min, held at
100% from 10 to 32 min, and then dropped back to 25% from 32 to 37 min, with a 3 min post
run time. During each sample run, reference ions were continuously monitored to provide
accurate mass corrections (purine and HP-0921 acetate adduct, Agilent G1969-85001).
Typically, the instrument exhibited 2 ppm mass accuracy. Tandem MS was conducted by
targeting ions under collision-induced dissociation (CID) to determine parent ion structures.
Agilent software Mass hunter was used for data acquisition (Version B05) and for further data
analysis (Qualitative and Quantitative Analysis, Version B07). The mass accuracy for compound



identification and quantification was set at ± 10 ppm. Calibration curves were generated from
diluted standard compound mixtures. Recoveries of the extraction and quantification were
performed by spiking known amounts of standards to blank QF filters. Then the spiked blank
filters were extracted and analyzed the same way as ambient collected samples. The average
recoveries of standard compounds are listed in supplemental information Table S5 and ranged
from $75.2 \pm 5.6$ to $129.4 \pm 4.2\%$. Isomers were identified for several compounds, no further
separation was conducted and combined total concentrations are reported in this study.

2.4 Source apportionment by Positive Matrix Factorization

Positive Matrix Factorization (PMF) was used to identify potential sources of compounds

measured at Coweeta. Here we use the PMF2 model (Paatero, 1998a, b) coupled with a bootstrap
technique (Hemann et al., 2009), which has been applied in a number of previous studies (Xie et
al., 2012, 2013, 2014,). Briefly, PMF resolves factor profiles and contributions from a series of
PM compositional data with an uncertainty-weighted least-squares fitting approach; the coupled
stationary bootstrap technique generates 1000 replicated data sets from the original data set and
each was analyzed with PMF. Normalized factor profiles were compared between the base case
solution and bootstrapped solutions, so as to generate a factor matching rate. The determination
of the factor number was based on the interpretability of different PMF solutions (3-6 factors)
and factor matching rate (>50%). Detailed data selection criteria are presented in supplemental
information.

**3.   Results and discussion**
3.1 Meteorology, NOx, and $O_3$

Statistics of atmospheric chemistry and meteorological measurements are summarized by

season in Table 1.  In general, the sampling site was humid and cool, even in the summer, with
an average summer temperature of ~21°C and RH of 82%. During the fall, much lower
temperature (~12 °C) and less humid conditions (RH=78%) were observed. NOx concentrations
were generally less than 1ppb, which is considered typical for such a remote forest site removed
from major emission sources.

[$O_3$] was generally low (Table 1) with seasonal averages of 15 ppb to 25 ppb. Historical

seasonal [$O_3$] over the past 5 years (2011 to 2015) are shown in supplemental information Figure



S3. A spring maximum in [$O_3$] is typically observed at this site, with lower concentrations during
summer. Seasonal clustered back trajectories (Figure S4 in supplemental information) suggest
that during spring the Coweeta sampling site was under the influence from air masses transported
from Atlanta urban areas. In addition, a spring maximum [$O_3$] may be due to higher chemical
consumption of $O_3$ by reactive monoterpenes and sesquiterpene emitted in the forest.   With
observed relatively moderate summer temperatures and generally low [NOx], the site also
experiences frequent cloud cover in summer lowering the intensity of solar radiation which may
suppress ozone production relative to spring conditions. Additionally, deposition of $O_3$ to the
forest would be expected to peak during the summer, when leaf area is greatest. $O_3$ correlates
positively with NOx in summer and fall but not spring, indicating $O_3$ production might be
relatively more VOC-limited in spring than the other seasons in this region.

3.2 Bulk water soluble organic nitrogen and carbon

Water soluble bulk organic N (WSON) was estimated as the difference between WSTN

and the sum of the inorganic N species ($NH_4^+$, $NO_3^-$ and $NO_2^-$). Nitrogen component
contributions to WSTN are presented in Figure 1a, which shows $NH_4^+$ as the most abundant
component, contributing 85±11% w/w to total WSTN mass. The oxidized inorganic N
components ($NO_3^-$ and $NO_2^-$) accounted for less than 2% w/w of WSTN measured. Such a small
contribution of $NO_3^-$ to inorganic N (typically <10% of inorganic N ($NO_3^-$+$NH_4^+$)) in $PM_{2.5}$ is
consistent with long term CASTNET measurements at Coweeta. The average contribution of
WSON to WSTN over the entire study period was 14±11% w/w. This fraction reached a
maximum of ~18% w/w in the spring (average) and minimum of ~10% in the fall (average),
exhibiting pronounced seasonal variability. Within individual samples (Figure 1b), values ranged
from near zero to 45%.  Our study wide average of 14% falls within the range of measurements
at North American forest sites, including Duke Forest, North Carolina (~33%, Lin et al., 2010)
and Rocky Mountain National Park (14-21%) (Benedict et al., 2012).

WSOC accounted for roughly 62±13% of OC throughout the entire study period with no

significant seasonal variability.  A time series of OC and WSOC along with temperature and
precipitation is presented in Figure 1c. On average, OC concentrations increased during warmer
spring and summer seasons and decreased when the temperature decreased in fall.
Concentrations of OC were positively correlated with temperature (r=0.30, p<0.05), presumably



345 in response to emissions of biogenic precursors and formation of secondary organic aerosols by

346 photooxidation. Spring and summer were generally moist and warm with frequent precipitation

347 (relative humidity presented in Table 1). Precipitation events corresponded to decreasing OC and

348 WSOC concentrations demonstrating the scavenging effect due to wet deposition.

349  Spearman rank correlation coefficients among measured species and meteorological

350 variables as well as other gas phase measurements are presented in Table 2 for each season

351 ($p<0.01$ for values in bold). As expected, $NH_4^+$ and $SO_4^{2-}$ tracked well over each season (r>0.9,

352 $p<0.01$). $NH_4^+$ was mainly associated with $SO_4^{2-}$ given the fact that $NO_3^-$ and $NO_2^-$ were

353 generally negligible compared to $SO_4^{2-}$. WSOC is often used as an SOA surrogate and accounts

354 for a significant portion (62% w/w) of OC during all sampling periods. WSOC correlated

355 strongly with OC over both summer and fall (r>0.95, p<0.01), but less so during spring (r=0.74,

356 p<0.01). WSOC also positively correlates with WSON over spring and fall (r>0.75, p<0.01) but

357 less so during summer (r = 0.5, p > 0.01). Note that both [WSOC] and [OC] are highest in the

358 summer, which likely indicates higher biogenic emissions and SOA formation. However, the

359 weak WSON-WSOC correlation suggests a variety of source contributions over the different

360 seasons. [EC] was negligible over the entire study except a modest spike at the end of October

361 when wood burning was the most likely the source. Details of this event are discussed in the

362 subsequent sections.

365 3.3 Nitro-aromatics

366  Concentrations of nitro-aromatics, organosulfate/nitrooxy-organosulfate, and terpenoic

367 acids are summarized in Tables 3, S2, S3 and S4. A time series of compound class totals are

368 presented in Figure 2. Generally negligible concentrations of nitro-aromatics were observed

369 during spring and summer except for occasional spikes. However, higher concentrations of nitro-

370 aromatics were observed in the fall when moderate correlations were observed with levoglucosan

371 (Figure 3, r≥0.5, p<0.01; see table SI 6 for correlation coefficients). A residential wood burning

372 contribution is likely given the lower temperatures observed during this season. Similar positive

373 correlations between nitro-aromatics and wood burning are also reported during the winter

374 season (Gaston et al., 2016; Kahnt et al., 2013; Kitanovski et al., 2012; Iinuma et al., 2010,

375 2016). Smoke at the sampling site on October 19[th] and 21[st] coincided with firewood burning at



the main office of the Coweeta Hydrologic Laboratory, immediately adjacent to the sampling
location. Nitro-aromatics were relatively elevated, but no significant increase in organosulfates
or terpenoic acids were found from these fresh smoke events. In contrast, an example of an aged
biomass burning signal is illustrated on October 24th and 25th. Pronounced spikes of
nitrocatechol($C_6H_5NO_4$), methyl-nitrocatechol($C_7H_7NO_4$) and levoglucosan were observed
(Figure 3), along with elevated concentrations of organosulfates, OC and aged biogenic aerosol
tracers (terpenoic acids m/z 203 and 187 shown in Figure 4a, detailed discussion can be found in
the subsequent section). However, EC was only slightly higher.  This event did not correspond to
local burning at Coweeta and was most likely associated with long range transport.

Nitro-aromatics correlate with EC across the seasons; both are likely emitted from

biomass burning (Gaston et al., 2016; Iinuma et al., 2010; Kahnt et al., 2013; Mohr et al., 2013).
Interestingly, light absorption at λ= 365nm is highly correlated (r=0.80, p<0.01) with nitro-
aromatics in the fall when nitro-aromatic concentrations were elevated. In addition, NOx
correlates inversely (r=-0.72, p<0.01) with temperature in the fall. Lower fall temperatures in the
region may have resulted in frequent residential wood burning, which emits NOx and light
absorbing BrC (eg. nitro-aromatics) (Liu et al., 2015; Mohr et al., 2013). Although nitro-
aromatics account for a minor fraction of OM, they could potentially contribute to 4% of light
absorption by BrC (Mohr et al., 2013). Overall, nitro-aromatics displayed relatively week
correlation with WSON (r<0.65) across all seasons; the extreme low concentrations observed
suggest a generally small contribution of nitro-aromatics to WSON at the sampling site, hence
the lack of strong correlation.

3.4 Organosulfates and nitrooxy-organosulfates

Organosulfate concentrations were highest in summer and lowest in fall (Table 3), contributing
3.9 and 1.0 % w/w of organic matter (OM, estimated by applying an OM/OC factor of 2) mass,
respectively, during these seasons. Organosulfate formation is an example of heterogeneous
chemistry involving uptake of reactive precursors on acidified sulfate aerosols requiring a
mixture of biogenic and anthropogenic emissions. The air masses at Coweeta are mainly from
the southwest and westerly directions in spring and summer, but during fall may become more
stagnant and slow moving during southwesterly conditions or shift to the northwest (see
clustered back trajectories are shown in Figure S4). Because Atlanta, GA is southwest of





Coweeta, southwesterly flow during spring and summer may be associated with long range
transport of urban pollutants and precursors, including sulfate and sulfuric acid, leading to
elevated organosulfate formation compared to fall when the prevailing wind direction changes.
Among all organosulfates identified, the isoprene derived organosulfate (m/z 215, 2-
methyltetrol derived), which is formed from isoprene derived epoxydiols (IEPOX) under low
NOx conditions, was the most abundant; concentrations reached 167 ng/m$^3$ in summer. Similar
high concentrations were also reported in ambient samples collected at other sites in the
southeastern U.S. (Lin et al., 2013b; Worton et al., 2013). Of the six nitrooxy-organosulfates
identified, isoprene derived m/z 260 was most abundant, approximately 6-fold higher than
monoterpene derived m/z 294 nitrooxy-organosulfate.
A subset of possible organosulfates and nitrooxy-organosulfates produced from isoprene
and monoterpene oxidation exhibit strong correlations with distinctive SOA tracers (eg. markers
2-methylglyceric acid, C-5 alkene triols and methyltetrols for isoprene oxidation products; tracer
3-Hydroxyglutaric acid for pinene oxidation products) (see table SI 7). Lack of correlation
between nitrooxy-organosulfate m/z 294 and 3-hydroxyglutaric acid may indicate a nighttime
nitrate radical formation pathway rather than photochemical oxidation.  Given that NOx levels at
the rural Coweeta sampling site were typically less than 1ppb, photo-oxidation pathways
involving high [NOx] to form nitrooxy-organosulfates are not likely. Nighttime nitrate radical
chemistry is the most likely formation mechanism under such conditions. In contrast to our
observations, He et al. (2014) report good correlations (r>0.5, p<0.01) of m/z 294 with 3-
hydroxyglutaric acid and higher daytime m/z 294 concentrations for summer samples collected
in Pearl River Delta, China, where a seasonal average NOx level of 30 ppb was observed. The
authors suggested that the dominant m/z 294 formation pathway was through daytime
photochemistry rather than nighttime NO$_3$ chemistry. The extremely low NOx levels at our study
site compared to that measured by He et al. may explain the opposite behavior in terms of m/z
294 formation mechanisms.
Organosulfates exhibited statistically significant correlations with WSON only in the
summer (r=0.64, p<0.01), which reflected the importance of N containing organosulfates or their
formation chemistry to WSON.  During this season, nitrooxy-organosulfates accounted for ~2%
of bulk WSON, on average. A strong correlation may therefore not be expected.



3.5 Terpenoic acids
Terpenoic acids, which provide insight into the extent of biogenic compound oxidation
and atmospheric processing (i.e., aerosol aging), were the most abundant group of compounds
relative to nitro-aromatics and organosulfates. On average, terpenoic acids accounted for 6.5 to
8.7% w/w of OM in $PM_{2.5}$. The warmer spring and summer periods show higher production of
terpenoic acids compared to the cool and drier fall season. Higher emissions of biogenic VOC
precursors as well as higher solar radiation intensities during warm seasons, which drive
photochemistry, are factors contributing to observed seasonal variability.
The terpenoic acids correlate well with WSOC and OC (Table 2). This is expected as
terpenoic acids account for a substantial portion of OM at the site. Individual acids (except
compounds $C_7H_{10}O_4$ and $C_9H_{14}O_4$) exhibit strong correlations with the pinene derived SOA
tracer 3-hydroxyglutaric acid ($r>0.75$, $p<0.01$; correlation coefficients shown in the supplemental
information Table S8), indicating the presence of α-/β-pinene oxidation products. The poor
correlations between acids $C_7H_{10}O_4$ (m/z 157) and $C_9H_{14}O_4$ (m/z 185) suggests the presence of
biogenic VOC precursors other than α-/β-pinene, such as limonene and $\Delta^3$-carene (Gomez-
Gonzalez et al., 2012).
Recent chamber studies identified several terpenioc acid structures also observed in
ambient aerosol samples, including 3-methyl-1,2,3-butanetricarboxylic acid (MBTCA, $C_8H_{12}O_6$,
m/z 203), 2-hydroxyterpenylic acid ($C_8H_{12}O_5$, m/z 187), terpenylic acid ($C_8H_{12}O_4$, m/z 171) and
diaterpenylic acid acetate (DTAA, $C_{10}H_{16}O_6$, m/z 231) (Claeys et al., 2009; Kahnt et al., 2014).
MBTCA and 2- hydroxyterpenylic acid have been identified as highly oxygenated, higher
generation α-pinene SOA markers, and observed in high abundance in ambient aerosols (Gomez-
Gonzalez et al., 2012; Kahnt et al., 2014; Muller et al., 2012; Szmigielski et al., 2007).
Additionally, terpenylic acid and DTAA are characterized as early photooxidation products from
α-pinene ozonolysis.  Claeys et al. (2009) proposed further oxidation processes (aging) of
terpenylic acid involving OH radical chemistry to form 2- hydroxyterpenylic acid.  Figure 4
provides a time series of the terpenoic acids identified in this study. In general, 2-
hydroxyterpenylic acid was the most abundant species across the seasons. To assess the extent of
aging, concentration ratios of higher generation oxidation products ($C_8H_{12}O_6$, m/z 203 and
$C_8H_{12}O_5$, m/z 187) to early oxidation fresh SOA products ($C_8H_{12}O_4$, m/z 171 and $C_{10}H_{16}O_6$, m/z
231) are calculated.  Estimated seasonal averages of these ratios are 3.98, 4.37 and 2.44 for



spring, summer and fall, respectively. Thus, during spring and summer, aerosols observed at the
site were more aged. Figure 4 shows the correlation of these ratios with temperature (r=0.79,
p<0.001) and solar radiation (r=0.23, p<0.1). A clear relationship between temperature and OH
radical initiated oxidation (aging) is evident. However, oxidation appears less dependent on solar
radiation at our sampling site. Similar higher contribution of these aged biogenic SOA tracers
was also reported under warm summer conditions characterized by high temperature and high
solar radiation (Claeys et al., 2012; Gomez-Gonzalez et al., 2012; Hamilton et al., 2013; Kahnt et
al., 2014).

Terpenoic acids may also provide some insight into the formation mechanisms of
organosulfates. While organosulfate concentrations are highest during summer, correlations with
$SO_4^{2-}$ are strongest during spring and fall and weakest during summer. Conversely,
organosulfates and terpenoic acids correlate strongly (r=0.91. p<0.01) during summer.
Terpenoic acids are either first or second generation oxidation products from gas phase
monoterpenes; particulate $SO_4^{2-}$ abundance should not substantially influence the gas-particle
partitioning of terpenoic acids. The strong correlation between organosulfates and terpenoic
acids in summer suggests organosulfate formation is limited by monoterpene emissions rather
than $SO_4^{2-}$ availability while in the spring and fall (especially fall), organosulfate production may
be more limited by $SO_4^{2-}$. Degree of particle neutralization, calculated as the molar ratio of $NH_4^+$
to the sum of $SO_4^{2-}$ and $NO_3^-$, averaged 0.94, 0.98 and 0.94 for spring, summer and fall,
respectively. Neutralization being close to but less than unity implies that aerosols are slightly
acidic at the site. Chamber studies have illustrated that acidified $SO_4^{2-}$ could enhance
heterogeneous reactions to form SOA from isoprene and monoterpenes (Iinuma et al., 2009;
Surratt et al., 2007, 2010). Similar positive correlations observed at the Coweeta site were also
found between isoprene tracers including isoprene derived organosulfates and $SO_4^{2-}$ by Lin et al.
(2013b) at a rural site in the southeastern U.S. However, in contrast to chamber experiments, this
study and other ambient field measurements have not provided clear evidence of acidity
enhancement of organosulfate formation (He et al., 2014; Lin et al., 2013b; Worton et al., 2011),
indicating possible differences in exact mechanisms and processing to form these organosulfates
under atmospheric conditions relative to chamber studies. Recent mechanistic modeling
simulations by Budisulistiorini et al., (2017) suggest that the role of sulfate on IEPOX-





organosulfates formation might be through surface area uptake of IEPOX and rate of particle
phase reaction.

Very good correlations between WSON and terpenoic acids were observed during summer

and fall (r≥0.7, p<0.01). Given the secondary nature of terpenoic acids, this finding may suggest
that WSON during these two seasons is associated with more aged air masses and perhaps
dominated by secondary organic components rather than primary emitted N containing
constituents such as pollens, fungi and bacteria (Elbert et al., 2007; Miyazaki et al., 2014).

3.6 Contribution of identified N containing species to WSTN and WSON

Nitro-aromatics and nitrooxy-organosulfates were estimated to account for as much as

28% of WSON, which reflected the abundance and potential importance of these groups of
species to the atmospheric N deposition budget. Seasonal average ratios of identified WSON to
WSTN ranged from 1.0 to 4.4% with the highest recorded for fall (Table 4). Nitrooxy-
organosulfates dominated over nitro-aromatics as a source of organic nitrogen, contributing >
90% to identified WSON across seasons. However, during episodes of biomass burning, nitro-
aromatics contribute as much as 32% of identified WSON compounds.  The ratio of WSON to
WSOC was estimated to be 0.05, 0.04 and 0.02 for spring, summer and fall, which implies
organic N being most enriched during spring, reflecting a spring maximum in seasonal emissions
of Organic N from biological sources (e.g. pollens, spores, leave litter decomposition) combined
with smaller contributions from secondary atmospheric processes.  The observed N/C ratios in
this study were slightly lower than those reported for other forest sites (0.03-0.09) (Lin et al.,
2010; Miyazaki et al., 2014), which are not as remote and pristine as the forest site in this study.
Anthropogenic influences at the study sites described by Lin et al. (2010) and Miyazaki et al.
(2014) such as $[SO_4^{2-}]$ and [NOx] were ~ 5 times higher than those measured at the Coweeta site.
Concentration weighted average N/C ratios for identified compounds (nitro-aromatics,
organosulfates/nitrooxy-organosulfates and terpenoic acids) in this study were estimated to be
0.003. This value is 10 times less than the overall N/C ratio observed at the site, which indicates
existence of other higher N content species in the aerosols.

3.7 PMF analysis

PMF analysis was conducted to identify individual source contributions to total WSOC.

Factor profiles and time series of factor contributions are presented in figures 5 and 6. Listed in





order of percent contribution to WSOC, the five factors which were resolved include secondary
sulfate processing (35.3%), isoprene SOA (24.3%), WSON containing OM (20.0%), biomass
burning (15.1%) and monoterpene SOA (5.2%). Overall, these factors could explain 89±2% of
observed WSOC (r=0.88, p<0.0001). The secondary sulfate profile contained a signature of high
$SO_4^{2-}$, which was most likely present as fine particulate $(NH_4)_2 SO_4$ and $NH_4HSO_4$. Secondary
sulfate was the most important factor during spring, though was a significant contributor in
summer and fall as well.  Isoprene SOA, which was identified based on isoprene derived
organosulfates and isoprene SOA markers, was the most important factor during summer. The
biomass burning factor, which exhibited a high portion of nitro-aromatic and levoglucosan
markers, dominated in the fall. This pattern agreed well with observed patterns of nitro-aromatic
compounds. Monoterpene SOA, which was resolved based on the composition of monoterpene
derived organosulfates, was overall a minor contributor with the exception of a few samples
during the fall intensive.

WSON containing OM contributed 20% to WSOC, overall, demonstrating a significant

association between organic N and C in $PM_{2.5}$ at our study site. The WSON containing OM
source profile exhibited weak correlation with most measured species with the exception of
modest correlations with terpenoic acids. WSON containing OM contributed more to WSOC in
late spring and early summer, which was consistent with observed higher production of nitrooxy-
organosulfates during these sampling periods as well as terpenoic acids. The relationship with
terpenoic acids may reflect an association of WSON with more aged air masses. Because nitro-
aromatics and nitrooxy-organosulfates contribute only a small portion of WSON, on average, the
20% contribution of WSON containing OM to WSOC primarily reflects the contribution of
organic N present in bulk WSON but unspeciated in this work.

4.  **Conclusions**

Ambient $PM_{2.5}$ collected at a temperate mountainous forest site were investigated on a bulk

chemical and a molecular level during spring, summer, and fall of 2015. Analyses focused on
speciation of nitro-aromatics associated with biomass burning, organosulfates produced from
biogenic SOA precursors, and terpenoic acids formed from monoterpene oxidation. Among these
three groups, terpenoic acids were estimated to be most abundant, contributing up to a seasonal
average of 8.7% of OM in $PM_{2.5}$ during spring.  Warm periods in spring and summer exhibited



highest production of terpenoic acids, when SOA correspondingly showed a higher degree of
aging.  Relative abundance of aged biogenic SOA tracers (MBTCA and 2- hydroxyterpenylic
acid), which reflect the degree of organic aerosol aging, showed a strong correlation with
temperature.  Such a relationship might indicate temperature dependence of OH radical initiated
oxidation steps or aging in the formation of higher generation oxidation products.

Organosulfates showed a peak in summer and lowest concentrations during fall,

contributing averages of 3.9 and 1.0 % of OM mass, respectively, during these seasons. Isoprene
derived organosulfate (m/z 215, 2-methyltetrol derived), formed from isoprene derived
epoxydiols (IEPOX) under low NOx conditions, was the most abundant identified organosulfate
(up to 167 ng/m$^3$ in summer).  This observation is consistent with observations of low NOx
levels (< 1ppb on average) at our study site. Nighttime nitrate radical chemistry is most likely the
dominant formation mechanism for nitrooxy-organosulfates measured at this remote site with
background level NOx.

Nitro-aromatics were most abundant at our study site during the fall (up to 0.01% of OM

mass).  Moderate correlations were observed between nitro-aromatics and the biomass burning
marker levoglucosan, indicating a common origin. Nitro-aromatics also correlated well with EC
across seasons.  Highest concentrations of nitro-aromatics, specifically nitrocatechol and methyl-
nitrocatechol, were associated with aged biomass burning plumes as indicated by
correspondingly high concentrations of terpenoic acids.

Bulk measurements determined that WSOC accounted for 62±13% of OC throughout the

entire study period without significant seasonal variability. PMF analysis indicated that a
significant portion of this organic carbon was associated with a resolved factor of WSON
containing OM. As a component of total nitrogen in PM$_{2.5}$, largest contributions of WSON to
WSTN were observed in spring (~ 18% w/w) and lowest in the fall (~10% w/w). On average,
identified nitro-aromatic and nitrooxy-organosulfate compounds accounted for a small fraction
of WSON, ranging from ~ 1% in spring to ~ 4% in fall, though were observed to contribute as
much as 28% w/w of WSON in individual samples. Of the organic N compounds speciated in
this study, nitrooxy-organosulfates dominated over nitro-aromatics as a source of organic
nitrogen, contributing > 90% to WSON across seasons.  As a component of WSON, nitro-
aromatics were most important during episodes of biomass burning, when their contribution to
identified and total WSON was as much as 32% and 3%, respectively.  Concentration weighted



average N/C ratios for compounds identified in this study were estimated to be 0.003. This
number is an order of magnitude lower than the overall N/C ratio observed, indicating a
predominance of other uncharacterized N species. Other N containing substituents of WSON
could include amino acids, amines, urea and N-heterocyclic compounds as well as substances of
biological origin such as spores, pollens and bacteria (Cape et al., 2011; Neff et al., 2002a).
Ratios of WSON to WSOC indicate organic C being most enriched by organic N during spring,
perhaps reflecting a spring maximum in seasonal emissions of organic N from biological sources
combined with smaller contributions from secondary atmospheric processes (e.g., nitrooxy-
organosulfates).
Although nitro-aromatics and nitrooxy-organosulfates contribute a relatively small
fraction of organic N in $PM_{2.5}$ at our study site, our observations shed light on this complex but
largely unknown portion of the atmospheric N budget. Our results provide further understanding
of the patterns and composition of SOA in a remote mountain environment previously
uncharacterized. Similar to our results, other studies generally find that individual groups of
organic N compounds (e.g., amines, amino acids, urea) cannot explain the majority of bulk
WSON, (Cape et al., 2011; Day et al., 2010; Place et al., 2017; Samy et al., 2013), which
globally accounts for ~25% of total N in rainfall (Cape et al., 2011; Jickells et al., 2013). As
methodological advances allow for greater speciation of this large pool of atmospheric N, future
work should emphasize analysis of both primary and secondary forms of organic N in individual
samples, in addition to bulk analyses, so that a more complete picture of organic N composition
may be developed for specific atmospheric chemical and meteorological conditions.
Additionally, as progress is made in better characterizing the composition and sources of
atmospheric organic N, the ecological and atmospheric science communities must work together
to develop a better understanding of the role of atmospheric organic N in ecosystem N cycling.

**Supplemental Information available**

**Acknowledgements**

We would like to acknowledge Pamela Barfield, Ryan Daly, Aleksandra Djurkovic, David
Kirchgessner, John Offenberg, Bakul Patel and Bill Preston for laboratory and field support. We
also would like to thank Joshua G. Hemann and Michael P.Hannigan for the PMF source codes



and Christopher Oishi, Patsy Clinton and Chuck Marshall for assistance with meteorological data
sets. We would like to thank the U.S. Forest Service, Southern Research Station for the
opportunity to conduct this study at the Coweeta Hydrologic Laboratory and for the contribution
of meteorological data used in our analysis. We also thank internal EPA reviewers Chris Geron
and Havala Pye for their comments and suggestions. The views expressed in this article are those
of the authors and do not necessarily represent the views or policies of the U.S. EPA.   Mention
of trade names does not constitute endorsement or recommendation of a commercial product by
U.S. EPA.

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



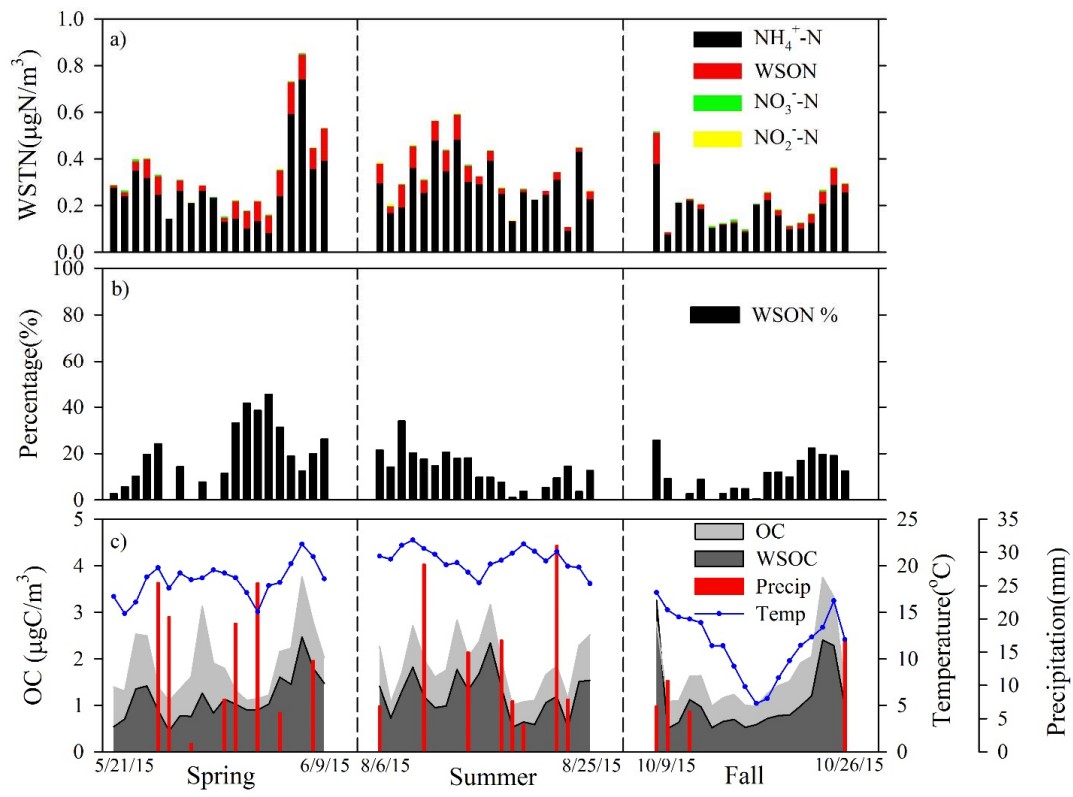

Figure 1. a) Individual concentrations of nitrogen components to WSTN ($NH_4^+$, $NO_3^-$, $NO_2^-$ and WSON); b) Percent contribution of WSON to WSTN; c) Time series of OC, WSOC, temperature and precipitation. Start and end dates of each intensive sampling periods are shown.





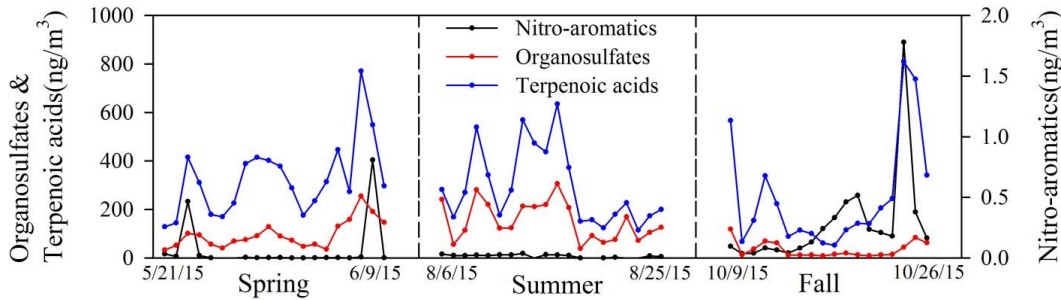

Figure 2. Time series of summed compound group concentrations of nitro-aromatics, organosulfates and terpenoic acids.




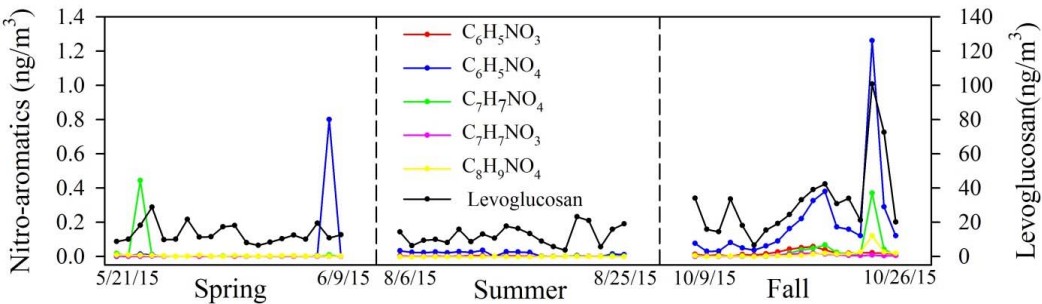

Figure 3. Time series of individual nitro-aromatics compounds as well as levoglucosan.



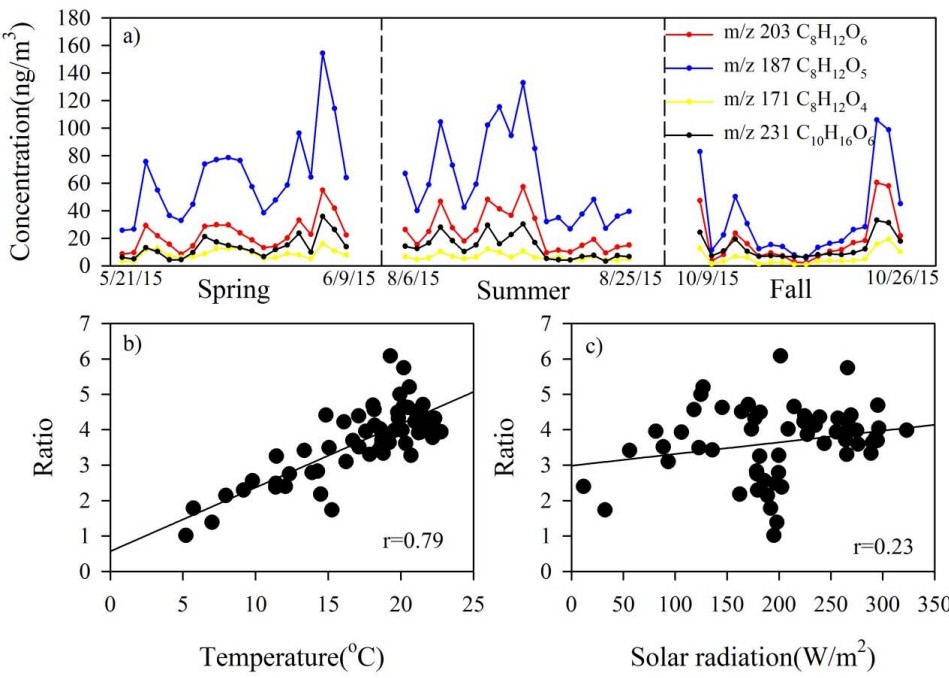

Figure 4. a) Time series of these four identified terpenoic acids(3-methyl-1,2,3-butanetricarboxylic acid(MBTCA, $C_8H_{12}O_6$, m/z 203), 2-hydroxyterpenylic acid($C_8H_{12}O_5$, m/z 187), terpenylic acid($C_8H_{12}O_4$, m/z 171) and Diaterpenylic acid acetate(DTAA, $C_{10}H_{16}O_6$, m/z 231)); b) correlation of concentration ratios of higher generation oxidation products( $C_8H_{12}O_6$, m/z 203 and $C_8H_{12}O_5$, m/z 187) to early oxidation fresh SOA products($C_8H_{12}O_4$, m/z 171 and $C_{10}H_{16}O_6$,m/z 231) with temperature and ; c) with solar radiation.



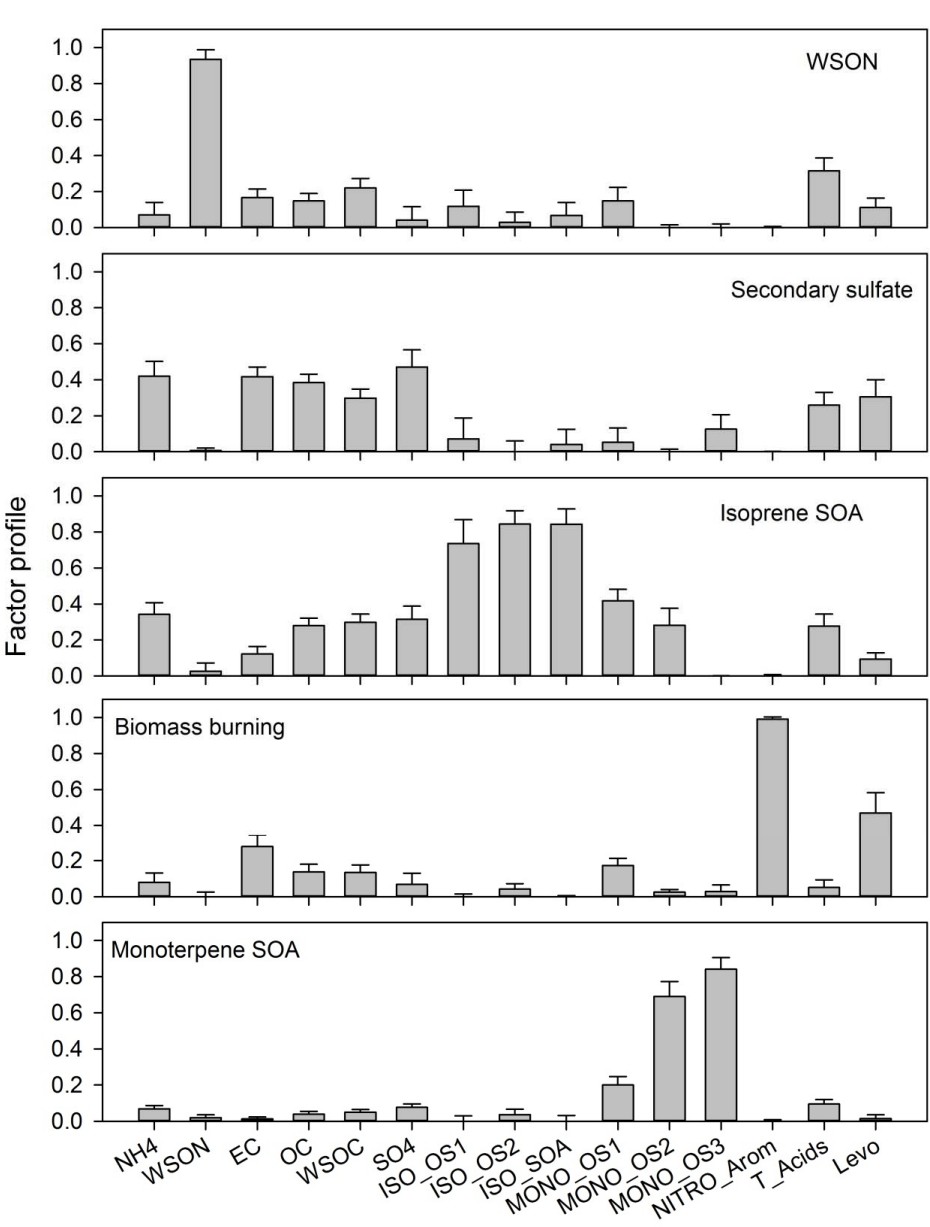

Figure 5. Normalized factor profiles (error bar represents one standard deviation).





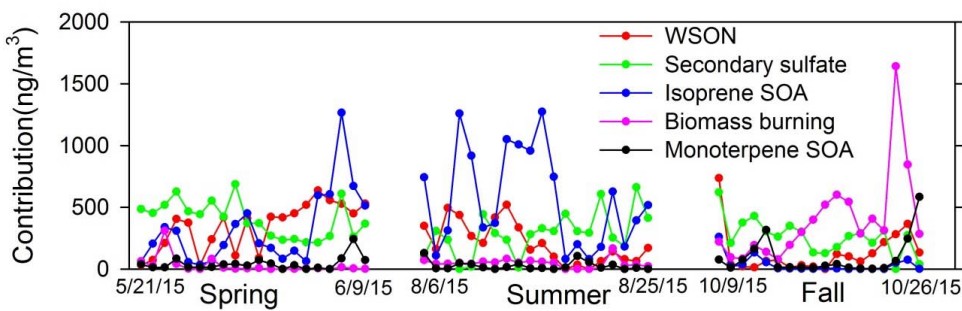

Figure 6. Time series of factor contributions to WSOC.





Table 1. Summary of particulate and gaseous species measured at Coweeta sampling site in 2015.

| (µg/m³) | Spring | | | | Summer | | | | Fall | | | |
|---|---|---|---|---|---|---|---|---|---|---|---|---|
| | mean | median | min | max | mean | median | min | max | mean | median | min | max |
| OM (OC*2) | 3.77 | 3.41 | 2.18 | 7.52 | 3.80 | 3.79 | 2.00 | 6.32 | 3.36 | 2.85 | 1.96 | 7.49 |
| EC | 0.05 | 0.05 | 0.03 | 0.10 | 0.05 | 0.05 | 0.02 | 0.08 | 0.07 | 0.07 | 0.03 | 3.75 |
| WSOC | 1.14 | 1.03 | 0.45 | 2.47 | 1.22 | 1.24 | 0.53 | 2.34 | 1.09 | 0.78 | 0.50 | 3.25 |
| WSTN | 0.33 | 0.29 | 0.14 | 0.86 | 0.34 | 0.32 | 0.11 | 0.59 | 0.21 | 0.20 | 0.08 | 0.52 |
| WSON | 0.06 | 0.07 | ND | 0.14 | 0.05 | 0.03 | ND | 0.11 | 0.03 | 0.02 | ND | 0.13 |
| $NH_4^+$-N | 0.27 | 0.24 | 0.08 | 0.74 | 0.29 | 0.28 | 0.09 | 0.48 | 0.18 | 0.17 | 0.08 | 0.38 |
| $NO_3^-$-N | 0.00 | 0.00 | ND | 0.01 | 0.00 | 0.00 | ND | 0.01 | 0.00 | 0.00 | ND | 0.01 |
| $NO_2^-$-N | 0.00 | 0.00 | ND | 0.00 | 0.00 | 0.00 | ND | 0.01 | 0.00 | 0.00 | ND | 0.00 |
| $SO_4^{2-}$ | 0.99 | 0.93 | 0.26 | 2.44 | 1.01 | 0.95 | 0.31 | 1.85 | 0.63 | 0.58 | 0.30 | 1.33 |
| $O_3$(ppb) | 25.1 | 21.6 | 13.9 | 46.1 | 15.8 | 15.8 | 9.0 | 22.8 | 19.4 | 20.5 | 11.1 | 26.9 |
| NOx(ppb) | 0.75 | 0.79 | 0.45 | 1.03 | 0.54 | 0.58 | 0.24 | 0.91 | 0.65 | 0.68 | 0.43 | 0.89 |
| Temp(°C) | 18.4 | 18.6 | 14.8 | 22.3 | 20.7 | 20.6 | 18.1 | 22.8 | 11.6 | 11.7 | 5.2 | 17.1 |
| RH% | 81.7 | 84.9 | 61.0 | 94.8 | 82.1 | 83.1 | 71.9 | 88.5 | 77.7 | 74.9 | 65.1 | 92.0 |
| Radiation | 235 | 265 | 81 | 296 | 205 | 201 | 106 | 323 | 151 | 180 | 12 | 203 |





Table 2. Spearman rank correlation coefficients among measured species and meteorological variables by season. Nitro-aromatics (Nitro), Organosulfates (OS), and terpenoic acids (Tacids) represent group summed concentrations.

| Spring | OC | WSOC | $NO_3^-$ | $NH_4$ | $SO_4^{2-}$ | WSON | $Abs_{365}$ | Nitro | OS | Tacids | $O_3$ | $NO_x$ | Temp | RH | radiation | Precip |
|---|---|---|---|---|---|---|---|---|---|---|---|---|---|---|---|---|
| EC | **0.853** | 0.474 | 0.177 | **0.690** | **0.705** | 0.129 | **0.875** | **0.583** | **0.645** | **0.579** | 0.430 | 0.263 | 0.364 | **-0.627** | 0.520 | -0.458 |
| OC | | **0.737** | 0.069 | **0.767** | **0.708** | 0.328 | **0.773** | 0.541 | **0.848** | **0.761** | 0.275 | 0.498 | 0.543 | -0.408 | 0.441 | -0.315 |
| WSOC | | | 0.105 | 0.523 | 0.429 | **0.768** | 0.424 | 0.241 | **0.805** | **0.723** | 0.185 | 0.543 | 0.472 | -0.059 | 0.135 | -0.145 |
| $NO_3^-$ | | | | 0.15 | 0.137 | 0.129 | 0.108 | 0.492 | -0.104 | -0.051 | 0.559 | 0.084 | -0.203 | -0.564 | 0.362 | -0.169 |
| $NH_4$ | | | | | **0.944** | 0.457 | **0.842** | 0.355 | **0.684** | 0.298 | 0.474 | 0.189 | 0.439 | -0.510 | 0.441 | -0.362 |
| $SO_4^{2-}$ | | | | | | 0.400 | **0.827** | 0.277 | **0.642** | 0.229 | 0.457 | 0.051 | 0.540 | -0.526 | 0.374 | -0.306 |
| WSON | | | | | | | 0.215 | -0.113 | 0.522 | 0.236 | 0.215 | 0.140 | 0.326 | 0.080 | -0.105 | 0.055 |
| $Abs_{365}$ | | | | | | | | 0.456 | **0.591** | 0.349 | 0.495 | 0.174 | 0.254 | **-0.612** | 0.507 | -0.529 |
| Nitro[1] | | | | | | | | | 0.278 | 0.426 | 0.493 | 0.319 | 0.021 | -0.537 | 0.307 | -0.177 |
| OS[2] | | | | | | | | | | **0.759** | 0.080 | 0.341 | **0.644** | -0.084 | 0.162 | -0.140 |
| Tacids[3] | | | | | | | | | | | -0.066 | **0.571** | 0.442 | 0.000 | 0.141 | -0.066 |
| $O_3$ | | | | | | | | | | | | 0.068 | 0.026 | **-0.797** | 0.453 | -0.219 |
| $NO_x$ | | | | | | | | | | | | | 0.227 | -0.068 | 0.257 | -0.165 |
| Temp | | | | | | | | | | | | | | -0.111 | 0.183 | 0.061 |
| RH | | | | | | | | | | | | | | | **-0.786** | 0.551 |
| Radiation | | | | | | | | | | | | | | | | **-0.734** |

[1]nitro-aromatics; [2]organosulfates; [3]terpenoic acids; values in bold indicate p<0.01

| Summer | OC | WSOC | $NO_3^-$ | $NH_4$ | $SO_4^{2-}$ | WSON | $Abs_{365}$ | Nitro | OS | Tacids | $O_3$ | $NO_x$ | Temp | RH | radiation | Precip |
|---|---|---|---|---|---|---|---|---|---|---|---|---|---|---|---|---|
| EC | **0.671** | **0.659** | 0.113 | **0.626** | 0.555 | **0.562** | 0.546 | **0.576** | 0.474 | 0.537 | 0.325 | 0.242 | -0.402 | -0.384 | 0.465 | -0.356 |
| OC | | **0.961** | 0.233 | **0.627** | 0.517 | 0.556 | 0.558 | 0.523 | **0.856** | **0.823** | 0.304 | 0.289 | -0.379 | -0.300 | 0.269 | -0.189 |
| WSOC | | | 0.263 | **0.592** | 0.490 | 0.549 | 0.397 | **0.564** | **0.820** | **0.835** | 0.247 | 0.238 | -0.302 | -0.325 | 0.259 | -0.269 |
| $NO_3^-$ | | | | 0.343 | 0.271 | 0.355 | -0.143 | 0.165 | 0.325 | 0.469 | **0.642** | **0.665** | 0.120 | -0.279 | 0.263 | 0.181 |
| $NH_4$ | | | | | **0.977** | 0.550 | 0.405 | 0.535 | **0.609** | **0.585** | 0.320 | 0.415 | -0.108 | -0.388 | 0.421 | -0.218 |
| $SO_4^{2-}$ | | | | | | 0.465 | 0.343 | 0.477 | 0.487 | 0.474 | 0.241 | 0.350 | -0.090 | -0.426 | 0.447 | -0.290 |
| WSON | | | | | | | 0.170 | **0.633** | **0.642** | **0.692** | **0.698** | 0.391 | 0.026 | **-0.637** | 0.555 | -0.201 |
| $Abs_{365}$ | | | | | | | | 0.086 | 0.423 | 0.278 | 0.149 | 0.140 | **-0.586** | 0.012 | 0.167 | 0.089 |
| Nitro[1] | | | | | | | | | **0.573** | **0.614** | 0.367 | 0.418 | -0.116 | -0.346 | 0.247 | -0.446 |
| OS[2] | | | | | | | | | | **0.905** | 0.338 | 0.472 | -0.080 | -0.175 | 0.098 | 0.087 |
| Tacids[3] | | | | | | | | | | | 0.432 | 0.531 | -0.150 | -0.263 | 0.138 | -0.035 |
| $O_3$ | | | | | | | | | | | | **0.621** | -0.045 | **-0.607** | **0.571** | -0.046 |
| $NO_x$ | | | | | | | | | | | | | -0.116 | -0.049 | 0.018 | 0.214 |
| Temp | | | | | | | | | | | | | | -0.097 | -0.012 | 0.172 |
| RH | | | | | | | | | | | | | | | **-0.919** | **0.607** |
| Radiation | | | | | | | | | | | | | | | | **-0.583** |





| Fall | OC | WSOC | NO$_3^-$ | NH$_4$ | SO$_4^{2-}$ | WSON | Abs$_{365}$ | Nitro | OS | Tacids | O$_3$ | NOx | Temp | RH | radiation | Precip |
|---|---|---|---|---|---|---|---|---|---|---|---|---|---|---|---|---|
| EC | **0.719** | **0.695** | 0.449 | 0.216 | 0.127 | **0.707** | **0.897** | **0.779** | 0.154 | 0.472 | 0.042 | 0.106 | -0.036 | -0.044 | -0.100 | -0.380 |
| OC | | **0.955** | 0.077 | 0.434 | 0.333 | **0.837** | **0.715** | 0.340 | 0.554 | **0.897** | -0.282 | -0.189 | 0.525 | 0.441 | -0.441 | 0.047 |
| WSOC | | | 0.092 | **0.593** | 0.494 | **0.816** | **0.668** | **0.362** | **0.649** | **0.922** | -0.222 | -0.152 | 0.474 | 0.422 | -0.470 | 0.146 |
| NO$_3^-$ | | | | -0.044 | -0.053 | 0.106 | 0.385 | 0.445 | -0.300 | -0.088 | 0.257 | 0.084 | -0.375 | -0.461 | 0.265 | -0.385 |
| NH$_4$ | | | | | **0.983** | 0.490 | 0.191 | 0.209 | **0.874** | **0.664** | -0.158 | -0.096 | 0.356 | 0.350 | -0.410 | 0.265 |
| SO$_4^{2-}$ | | | | | | 0.399 | 0.100 | 0.152 | **0.833** | 0.571 | -0.110 | -0.086 | 0.313 | 0.290 | -0.342 | 0.244 |
| WSON | | | | | | | **0.789** | 0.486 | 0.546 | **0.746** | -0.143 | 0.036 | 0.364 | 0.441 | -0.538 | 0.224 |
| Abs$_{365}$ | | | | | | | | **0.802** | 0.110 | 0.494 | 0.150 | 0.286 | -0.096 | 0.011 | -0.226 | -0.273 |
| Nitro | | | | | | | | | 0.001 | 0.187 | 0.313 | 0.445 | -0.455 | -0.226 | 0.009 | -0.378 |
| OS | | | | | | | | | | **0.746** | -0.350 | -0.356 | **0.659** | 0.573 | -0.581 | 0.466 |
| Tacids | | | | | | | | | | | -0.401 | -0.249 | **0.653** | **0.628** | -0.587 | 0.241 |
| O$_3$ | | | | | | | | | | | | **0.664** | -0.746 | -0.820 | 0.602 | -0.340 |
| NOx | | | | | | | | | | | | | **-0.719** | -0.418 | 0.389 | -0.303 |
| Temp | | | | | | | | | | | | | | **0.787** | -0.639 | 0.490 |
| RH | | | | | | | | | | | | | | | **-0.847** | **0.638** |
| Radiation | | | | | | | | | | | | | | | | **-0.640** |

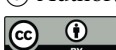



Table 3. Seasonal statistics of measured groups of compounds.

| (ng/m³) | Spring | | | | Summer | | | | Fall | | | |
|---|---|---|---|---|---|---|---|---|---|---|---|---|
| | mean | median | min | max | mean | median | min | max | mean | median | min | max |
| Nitro-aromatics | 0.07 | 0.00 | ND | 0.81 | 0.02 | 0.02 | ND | 0.04 | 0.28 | 0.17 | 0.04 | 1.78 |
| Organo-sulfates¹ | 96.77 | 83.05 | 33.07 | 255.17 | 153.36 | 125.41 | 38.93 | 306.66 | 34.69 | 15.27 | 0.17 | 118.68 |
| Terpenoic acids | 325.62 | 304.05 | 128.68 | 771.16 | 294.01 | 249.19 | 115.08 | 634.99 | 250.66 | 148.91 | 52.94 | 809.46 |
| % of OM² | | | | | | | | | | | | |
| %Nitro-aromatics | 0.00 | 0.00 | ND | 0.02 | 0.00 | 0.00 | ND | 0.00 | 0.01 | 0.01 | 0.00 | 0.02 |
| %Organo-sulfates | 2.47 | 2.42 | 1.19 | 3.64 | 3.87 | 3.80 | 1.95 | 5.56 | 0.98 | 0.63 | 0.31 | 2.21 |
| % Terpenoic acids | 8.65 | 8.29 | 4.62 | 12.88 | 7.50 | 7.77 | 3.80 | 11.64 | 6.48 | 5.21 | 2.70 | 12.00 |

¹ including nitrooxy-organosulfates; ²Fraction of each group of identified compounds (combined total) to organic matter





Table 4. Ratios of identified nitrogen containing compounds (nitro-aromatics and nitrooxy-organosulfates) to WSON.

| (ngN/m³) | Spring | | | | Summer | | | | Fall | | | |
|---|---|---|---|---|---|---|---|---|---|---|---|---|
| | mean | median | min | max | mean | median | min | max | mean | median | min | max |
| WSON | 59 | 74 | ND | 140 | 46 | 33 | ND | 105 | 25 | 15 | ND | 133 |
| Identified ON | 0.48 | 0.36 | 0.1 | 1.75 | 0.65 | 0.53 | 0.12 | 1.83 | 0.46 | 0.26 | 0.07 | 1.70 |
| Identified ON/WSON % | 1.02 | 0.64 | ND | 3.09 | 2.04 | 1.71 | ND | 7.84 | 4.37 | 1.50 | ND | 27.90 |