# Peer review of "Characterization of organic nitrogen in aerosols at a forest site in the southern Appalachian Mountains"

_Atmospheric Chemistry and Physics, 2018_

## Referee Comment (RC1) · Anonymous Referee #1 · 7 Feb 2018

Review of Chen et al I am reviewing this paper as someone with quite a lot of experience of measuring bulk organic nitrogen but with much less expertise in organic matter characterisation. Overall I think this is a useful paper that demonstrates the potential importance of the organic nitrogen in this region and also provides some useful characterisation of the some of the organic matter in the aerosol. The provision of data on organic carbon and nitrogen together is for me particularly useful. The wide range of data does allow some inter-component relationships to be used to suggest something about organic C and N cycling, although all the correlations may not prove a causal link. The sampling and analysis is state-of-the-art for the compounds analysed and provides a high quality and useful data set. I am happy to see it published but I would

suggest a few changes before publication.

Specific comments I do wonder if the title is really appropriate given how little of the organic nitrogen is characterised. Line 69 There is now a global model of atmospheric organic nitrogen cycling that should perhaps be referenced – Kanakidou et al 2012 Global Biogeochemical Cycles doi 10.1029/2011GB004277. Line 72 To my mind the work of Altieri cited here and their more recent paper (Altieri et al 2012 ACP 12 355703571) represent the best effort to characterise the atmospheric organic nitrogen and yet neither here or later in the paper is this work discussed. It is relevant because it identifies reduced nitrogen as a dominant component of the atmospheric organic nitrogen, yet the authors here are characterising oxidised nitrogen based organic matter. The rationale for their choice of compounds is not really explained in the section line 112-125 where I might expect it to be. Line 140 The site map needs to be in the main text not the supplementary material. Line 145-148 Given their importance from the results at this site, the authors might want to comment on ammonia sources. Line 151 – how many samples in total? I guess about 60 but it does help to know when looking at the statistical work. Line 151-3 Gonzalez-Benitez discussed the issue of semi-volatile organic nitrogen and it may be useful to at least note this, although it is very hard for most of us to sample for this. Line 221 I think "less than" should be "better than" if I understand the point Section 2.4 Please explain what the PMF is being used to investigate. The section here is a detailed description of the mathematical manipulations but it does not explain anything about the process to the non specialist. Line 317-8 How does how ozone consumption lead to a seasonal maximum? Line 337-340 For a wider audience I would suggest it is worth noting this %organic N is consistent with other data from the world beyond the USA. Line 342-344 The claimed seasonal cycle looks very small to me from the graphs. Line 349-352. The correlations are presented for each season, and that is OK although with only about 20 samples and so many variables I wonder about the statistical validity of the approach. I would therefore suggest that the equivalent correlation for the whole data set should also be presented. The observation of the correlation of WSON and WSOC is interesting and there is rather limited

such data valuable. I also note a much stronger correlation of WSON and NH4 than NO3. This is consistent with other data (see Cape et al 2011 cited) and points along with the Altieri work above, to a key role for reduced nitrogen in WSON formation. Line 359 "source contributions" of what? presumably WSON and C Line 374-7 We have all had problems such as described here, but is it really useful to include the samples collected when local burning impacted the sampler? This is particularly relevant because throughout much of the paper the authors show they can only characterise a few percent of the WSON. Then suddenly on line 508 they say they can characterise 28% which would be very impressive but I think this is for these local burning episodes and so by including this high percentage the authors may mislead readers into thinking as a community we are beginning to be able to characterise quite a lot of the WSON. This is also relevant to line 587 and the abstract. As the authors note in line 552 they and the rest of us have yet to be able to characterise very much of this material Line 434-5 Given how small a percentage of WSON appears to be made up of N containing organosulphate compounds, I'm not sure its correct to make the statement "which reflected... to WSON" here. Line 440 group of ORGANIC compounds Line 447 is 6-9% (which is what I think your report) really "a substantial proportion"? Line 446-453 Here and elsewhere I think the authors need to be careful about interpreting correlations as showing causal links. Line 562-565 I think the authors conclusions are valid for the material they have characterised, but that does not necessarily mean that all of the organic aerosol has been similarly aged. Line 581-3 I do not understand what the sentence starting "PMF analysis" means. I am not really sure that figure 5 and 6 add much to manuscript

---

## Referee Comment (RC2) · Anonymous Referee #2 · 5 Mar 2018

General comments

This paper presents the analytical results of water-soluble organic nitrogen (WSON) for both bulk and related molecular compounds in PM 2.5 filter samples collected at a remote montane forest site in the U.S. The authors present the season variation of WSON and related organic molecular compounds to characterize aerosol WSON and investigate its possible sources. Combination of bulk WSON and molecular tracer compounds related to WSON and WSOC obtained in the forest environment provides new insights into our understanding on aerosol WSON particularly from terrestrial biogenic sources. While the data presented are valuable, there are some important issues that

need to be worked out and clarified before I recommend its publication in ACP.

Specific comments

(1) One of my concern is on the interpretation for the positive correlation between biogenic SOA tracers and ambient temperature (Lines 470-476, 562-565). The authors conclude that such a relationship indicates temperature dependence of "oxidation." It may be true to some extents, but how about the temperature dependence of VOC emissions? Most of terpenes generally show temperature dependence of emission, which can also explain the correlation shown in this manuscript.

(2) The authors use the term "aged biogenic SOA" (e.g., Lines. 560-565 and others) in the text. Please add more discussion about specific time scale on this aging (hours, days?). This should be discussed relative to the time scale of transport (e.g., vertical mixing within the forest canopy or between the canopy and the above atmosphere, horizontal transport, etc.).

(3) Nitro-aromatics: In section 3.3., the authors conclude that the contribution of nitro-aromatics to WSON was generally "small," whereas they state potential importance of nitro-aromatics to the atmospheric N deposition budget (L.507-509) in section 3.6. These statements do not seem to be consistent and confusing.

(4) Lines 383-384: If the event cannot be attributed to local burning, then what is the most likely origin (source location)? "Long range transport" is not enough to explain the source of the observed particles in this event.

(5) Section 2.3: The authors should describe the measurement uncertainties for each analysis. This is particularly important for the analysis of WSON, whose measurement uncertainty includes propagation of errors of WSTN, $NO_3^-$, $NH_4^+$,....

(6) Figure 6: I think that the author should show time series of integrated factor contributions vs. the measured WSOC concentrations to show how well the PMF reproduced the measurements. Then the authors should show fractional contribution of each factor

to WSOC in the time series as they discuss it in the text.

(7) The authors use the term "N/C ratio" in the manuscript: Lines 42,43, 517, 524, 592, and 593. Should this term be "(WS)ON/OC ratio?" "N/C" includes inorganic N and elemental C.

(8) Lines 522-525: The identified-ON/WSON ratios also show a seasonal difference (Table 4). Can the authors add a few more statement on this difference in terms of unidentified compounds?

Minor comments

(9) Abstract: The authors should specify that the sampled aerosols are PM2.5.

(10) L.312: Please define "[O3]" here.

(11) L.394: Correct "extreme" to "extremely."

---

## Author Comment (AC1) · 9 Apr 2018

Point-to-point reply to reviewer 1's comments:

Review of Chen et al I am reviewing this paper as someone with quite a lot of experience of measuring bulk organic nitrogen but with much less expertise in organic matter characterisation. Overall I think this is a useful paper that demonstrates the potential importance of the organic nitrogen in this region and also provides some useful characterization of the some of the organic matter in the aerosol. The provision of data on organic carbon and nitrogen together is for me particularly useful. The wide range of data does allow some inter-component relationships to be used to suggest something

about organic C and N cycling, although all the correlations may not prove a causal link. The sampling and analysis is state-of-the-art for the compounds analysed and provides a high quality and useful data set. I am happy to see it published but I would suggest a few changes before publication.

Specific comments

Comment: I do wonder if the title is really appropriate given how little of the organic nitrogen is characterised.

Response: One very important purpose of our study is to provide more information on N- containing species in aerosols in forest environments. Although the identified and quantified speciated N-containing organic compounds only contributed a small fraction of the total WSON, which is consistent with other studies, we feel our resultsadd to the current scientific understanding of the issue and emphasizes the need to further characterize aerosol organic N composition in the atmosphere. We think the current title is appropriate considering our aim and purpose.

Comment: Line 69 There is now a global model of atmospheric organic nitrogen cycling that should perhaps be referenced – Kanakidou et al 2012 Global Biogeochemical Cycles doi 10.1029/2011GB004277.

Response: The reference by Kanakidou et al 2012 is now included in the text.

Comment: Line 72 To my mind the work of Altieri cited here and their more recent paper (Altieri et al 2012 ACP 12 355703571) represent the best effort to characterise the atmospheric organic nitrogen and yet neither here or later in the paper is this work discussed. It is relevant because it identifies reduced nitrogen as a dominant component of the atmospheric organic nitrogen, yet the authors here are characterising oxidised nitrogen based organic matter. The rationale for their choice of compounds is not really explained in the section line 112-125 where I might expect it to be.

Response: The reference by Altieri et al 2012 is now included in the Introduction section. The rationale for characterizing oxidized organic nitrogen is based on the possible sources of organic aerosols in a remote forest site. Emissions of the most abundant biogenic precursors, including isoprene, monoterpenes and sesquiterpnenes, from the forest are expected to favor the formation of oxidized low volatility compounds to nucleate or condense on to the particle phase. Also, the possibility of wildfires, which are believed to be a significant contributor to atmospheric OM loadings, is considered a potential source to our remote forest site.

Comment: Line 140 The site map needs to be in the main text not the supplementary material.

Response: The site map is showing the relative location of the sampling site as well as the whole research forest lab to the nearby roads and towns. Although it is providing useful information to understand the sources of the pollutants, we do not think the standalone site map will provide as much important information as other results figures and tables in the main text. We prefer to keep it in the supplemental section.

Comment: Line 145-148 Given their importance from the results at this site, the authors might want to comment on ammonia sources.

Response: A brief description of local emission sources is provided at Line 141 "Typical rural development is present to the east of the site, consisting of houses and small scale farming for hay and crop production including some scattered cow and horse pastures, which are small local ammonia emission sources." And also at Line 338 "Nitrogen component contributions to WSTN are presented in Figure 1a, which shows $NH_4^+$ as the most abundant component, contributing $85\pm11\%$ w/w to total WSTN mass. Typical $NH_4^+$ concentrations at the site were below 1.0 $\mu$g/m3(with an average of 0.32 $\mu$g/m3), which is expected for such a remote site with no major local and regional ammonia sources."

Comment: Line 151 – how many samples in total? I guess about 60 but it does help to know when looking at the statistical work.

Response: The total number of samples collected is 58. This information is provided on line 156 in the main text.

Comment: Line 151-3 Gonzalez-Benitez discussed the issue of semi-volatile organic nitrogen and it may be useful to at least note this, although it is very hard for most of us to sample for this.

Response: The following sentence is added to the text regarding difficulty in sampling for semi-volatile organic nitrogen: "Under some conditions, the 24hr integrated filter sampling technique may not fully retain all semi-volatile organic nitrogen compounds (Gonzalez Benitez et al., 2009)."

Comment: Line 221 I think "less than" should be "better than" if I understand the point

Response: "less than" is changed to "better than".

Comment: Section 2.4 Please explain what the PMF is being used to investigate. The section here is a detailed description of the mathematical manipulations but it does not explain anything about the process to the non specialist.

Response: The purpose and basic details of the PMF analysis are stated in the text as "Positive Matrix Factorization (PMF) was used to identify potential sources of compounds measured at Coweeta. Briefly, PMF resolves factor profiles and contributions from a series of PM compositional data with an uncertainty-weighted least-squares fitting approach. "

Comment: Line 317-8 How does how ozone consumption lead to a seasonal maximum?

Response: The sentence is revised as "In addition, a spring maximum [O3] may be due to higher chemical consumption of O3 by reactive monoterpenes and sesquiterpene emitted in the forest during summer."

Comment: Line 337-340 For a wider audience I would suggest it is worth noting this

[Figure]

%organic N is consistent with other data from the world beyond the USA.

Response: The following sentence is added "Moreover, the observed WSON contribution to WSTN in particles at Coweetais consistent with a global estimated range of 10-39% (Cape et al., 2011)."

Comment: Line 342-344 The claimed seasonal cycle looks very small to me from the graphs.

Response: The differences of OC during spring and summer compared to fall are discernible, but do not reflect a dramatic seasonal cycle..

Comment: Line 349-352. The correlations are presented for each season, and that is OK although with only about 20 samples and so many variables I wonder about the statistical validity of the approach. I would therefore suggest that the equivalent correlation for the whole data set should also be presented. The observation of the correlation of WSON and WSOC is interesting and there is rather limited such data valuable. I also note a much stronger correlation of WSON and NH4 than NO3. This is consistent with other data (see Cape et al 2011 cited) and points along with the Altieri work above, to a key role for reduced nitrogen in WSON formation.

Response: The correlations for the entire dataset are now included in Table 2. We appreciate the comment regarding "a much stronger correlation of WSON and NH4 than NO3". The following statement is now included at line 368 "It is also noted that a stronger correlation of WSON with NH4+ thanwith NO3- was observed, which might suggest a key role of reduced nitrogen in WSON formation (Cape et al., 2011; Jickells et al., 2013)".

Comment: Line 359 "source contributions" of what? presumably WSON and C

Response: Yes. The sentence is revised as "However, the weak WSON-WSOC correlation suggests a variety of source contributions to WSON and WSOC over the different seasons."

Comment: Line 374-7 We have all had problems such as described here, but is it really useful to include the samples collected when local burning impacted the sampler? This is particularly relevant because throughout much of the paper the authors show they can only characterise a few percent of the WSON. Then suddenly on line 508 they say they can characterize 28% which would be very impressive but I think this is for these local burning episodes and so by including this high percentage the authors may mislead readers into thinking as a community we are beginning to be able to characterise quite a lot of the WSON. This is also relevant to line 587 and the abstract. As the authors note in line 552 they and the rest of us have yet to be able to characterise very much of this material

Response: Yes, the particular sample characterized as 28% of WSON was impacted by local biomass burning. We agree that this was a special event and not typical of the other samples. However, we think it is relevant to be included as an example of the impact of local burning (i.e., "fresh" smoke). In order to avoid misleading the readers, we have revised the statement regarding this 28% characterized WSON: "On average, identified nitro-aromatic and nitrooxy-organosulfate compounds accounted for a small fraction of WSON, ranging from ïA¿ 1% in spring to ïA¿ 4% in fall, though were observed to contribute as much as 28% w/w of WSON in individual samples which were impacted by local biomass burning."

Comment: Line434-5 Given how small a percentage of WSON appears to be made up of N containing organosulphate compounds, I'm not sure its correct to make the statement "which reflected: : : to WSON" here.

Response: The emphasis of this statement is the importance of nitrogen containing organosulfate in summer to WSON relative to the other seasons based on the significant correlations observed. The statement is now revised to "Organosulfates exhibited statistically significant correlations with WSON only in the summer (r=0.64, p<0.01), which reflected the importance of N containing organosulfates or their formation chemistry to WSON during summer compared to the other seasons."

[Figure]

Comment: Line 440 group of ORGANIC compounds Line 447 is 6-9% (which is what I think your report) really "a substantial proportion"?

Response: Composition of atmospheric particulate OM is expected to be complex and could comprise hundreds and thousands of individual compounds. A single class of compounds with a particular functional group which contributes up to ∼10% of overall OM loading is substantial considering the complexity and numbers of potential component groups.

Comment: Line 446-453Here and elsewhere I think the authors need to be careful about interpreting correlations as showing causal links.

Response: We agree and appreciate the comment.

Comment: Line 562-565 I think the authors conclusions are valid for the material they have characterised, but that does not necessarily mean that all of the organic aerosol has been similarly aged.

Response: Thanks for the comment. We agree. The biogenic SOA tracers are most likely to reflect aging of the SOA portion of the organic aerosols, which was the intent of our statement in the text: "Warm periods in spring and summer exhibited highest production of terpenoic acids, when SOA correspondingly showed a higher degree of aging."

Comment: Line 581-3 I do not understand what the sentence starting "PMF analysis" means. I am not really sure that figure 5 and 6 add much to manuscript

Response: One important piece of information the PMF analysis adds to the study is the contribution of resolved WSON containing OM contributed 20% to WSOC, demonstrating a significant portion of OC is nitrogenated in PM2.5 at our study site. While our speciated analysis only identified a small fraction of the nitrogenated OC, there is definitely a need to conduct more in depth research to unveil a complete picture of organic N composition. We feel the results of the PMF analysis shown in Figures 5 and

6 provide useful information to readers.

---

## Author Comment (AC2) · 9 Apr 2018

Point to point reply to reviewer 2's comments: This paper presents the analytical results of water-soluble organic nitrogen (WSON) for both bulk and related molecular compounds in PM 2.5 filter samples collected at a remote montane forest site in the U.S. The authors present the season variation of WSON and related organic molecular compounds to characterize aerosol WSON and investigate its possible sources. Combination of bulk WSON and molecular tracer compounds related to WSON and WSOC obtained in the forest environment provides new insights into our understanding on aerosol WSON particularly from terrestrial biogenic sources. While the data presented

are valuable, there are some important issues that need to be worked out and clarified before I recommend its publication in ACP.

Specific comments Comment (1) One of my concern is on the interpretation for the positive correlation between biogenic SOA tracers and ambient temperature (Lines 470-476, 562-565). The authors conclude that such a relationship indicates temperature dependence of "oxidation." It may be true to some extents, but how about the temperature dependence of VOC emissions? Most of terpenes generally show temperature dependence of emission, which can also explain the correlation shown in this manuscript.

Response: The positive correlation referred to here is not between the abundance of the biogenic SOA tracers with temperature, rather we are using the relative abundance (concentration ratios) with temperature instead. It is stated in the text "To assess the extent of aging, concentration ratios of higher generation oxidation products ($C_8H_{12}O_6$, m/z 203 and $C_8H_{12}O_5$, m/z 187) to early oxidation fresh SOA products ($C_8H_{12}O_4$, m/z 171 and $C_{10}H_{16}O_6$, m/z 231) are calculated." We do agree that most terpene emissions are temperature dependent, but we think the relative abundance of higher generation products to early oxidation products probably will cancel out such effects.

Comment (2) The authors use the term "aged biogenic SOA" (e.g., Lines. 560-565 and others) in the text. Please add more discussion about specific time scale on this aging (hours, days?). This should be discussed relative to the time scale of transport (e.g., vertical mixing within the forest canopy or between the canopy and the above atmosphere, horizontal transport, etc.).

Response: Under atmospherically relevant conditions, the lifetime of $\alpha$-pinene SOA was reported as several days (4-7days) as a result of heterogeneous and condensed phase oxidation processes (Epstein et al., 2014). In addition, the precipitation frequency during spring and summer at the Coweeta site was quite high (as in days) and we think the aged biogenic SOA probably had a time scale of aging of days be-

fore scavenged by the precipitation. Figure 1C also shows such scavenging effect after precipitation on WSOC (a drop in WSOC right after precipitation occasions). A brief discussion on this topic is now added to the text at Line 495 "Based on typical chemical lifetime of biogenic SOA by OH oxidation and the precipitation frequency at the Coweeta site, biogenic SOA at Coweeta probably had an atmospheric lifetime of several days before depletion by oxidation processes (Epstein et al., 2014) and/or scavenging by precipitation."

Comment (3) Nitro-aromatics: In section 3.3., the authors conclude that the contribution of nitroaromatics to WSON was generally "small," whereas they state potential importance of nitro-aromatics to the atmospheric N deposition budget (L.507-509) in section 3.6. These statements do not seem to be consistent and confusing.

Response: L507-509 "Nitro-aromatics and nitrooxy-organosulfates were estimated to account for as much as 28% of WSON, which reflected the abundance and potential importance of these groups of species to the atmospheric N deposition budget." Here we are referring to the contribution of Nitro-aromatics and nitrooxy-organosulfates combined to N deposition budget, not nitro-aromatics alone. The sentence is now revised to "Nitro-aromatics and nitrooxy-organosulfates combined were estimated to account for as much as 28% of WSON, which reflected the abundance and potential importance of these groups of species to the atmospheric N deposition budget."

Comment (4) Lines 383-384: If the event cannot be attributed to local burning, then what is the most likely origin (source location)? "Long range transport" is not enough to explain the source of the observed particles in this event.

Response: The following analysis and discussion is now included at line 398: Clustering of backward trajectories (120hr duration for individual trajectories; 48 total trajectories covering the two-day event) suggests that northeast Georgia (shown in supplemental information Figure S5) is the most likely origin of the biomass burning event observed on October 24th and 25th.

[Figure]

Comment (5) Section 2.3: The authors should describe the measurement uncertainties for each analysis. This is particularly important for the analysis of WSON, whose measurement uncertainty includes propagation of errors of WSTN, NO3-, NH4+,: : :.

Response: Yes, the measurement uncertainty of WSON includes propagation of errors of WSTN, NO3-, NH4+ and NO2-. We have included the estimated measurement uncertainties in the text at Line 332: "The measurement uncertainty of WSON was estimated to be ïA¿ 30% from error propagation of WSTN (2%), NH4+ (1%), NO3- (1%) and NO2- (1%)."

Comment (6) Figure 6: I think that the author should show time series of integrated factor contributions vs. the measured WSOC concentrations to show how well the PMF reproduced the measurements. Then the authors should show fractional contribution of each factor to WSOC in the time series as they discuss it in the text.

Response: A linear regression plot for integrated factor contributions with measured WSOC is now included in the supplemental information as Figure S6. Linear regression coefficients are also provided in the figure. In Figure 6, the mean fractional contributions of each factor to WSOC are now included.

Comment (7) The authors use the term "N/C ratio" in the manuscript: Lines 42,43, 517, 524, 592,and 593. Should this term be "(WS)ON/OC ratio?" "N/C" includes inorganic N and elemental C.

Response: Yes, we are referring to WSON/WSOC ratio. We have revised the term "N/C ratio" throughout the manuscript to "WSON/WSOC ratio".

Comment (8) Lines 522-525: The identified-ON/WSON ratios also show a seasonal difference (Table 4). Can the authors add a few more statement on this difference in terms of unidentified compounds?

Response: The following discussion is included in the text regarding seasonal differences observed for identified-ON/WSON: "Moreover, identified ON/WSON ratio was

estimated to be 1.0, 2.0 and 4.4 for spring, summer and fall, respectively. Such differ­ences further suggest much more unidentified WSON compounds exist in spring when organic N was most enriched from biological processes."

Minor comments Comment (9) Abstract: The authors should specify that the sampled aerosols are PM2.5.

Response: Yes, PM2.5 is specified now in the abstract on line 24.

Comment (10) L.312: Please define "[O3]" here.

Response: [O3] definition is added as ozone concentration.

Comment (11) L.394: Correct "extreme" to "extremely."

Response: "extreme" is changed to "extremely".

---

## Author Response (AR2)

**Dear Authors:**

**Comment**: Thank you for your careful consideration of the referee reports. In general I find the responses satisfactory, but I have a few minor comments that I would like to see addressed prior to publication.

Response: Thank you for the suggestions and comments, we appreciate it.

**Comment**-line 372-373: I suggest moving the sentence "Details of this event..." before the previous sentence in order to improve readability.

**Response**: The sentence "Details of this event are discussed in the subsequent sections." is now moved before the previous sentence.

**Comment**-line 438: Organic nitrates can still form from photo-oxidation even at low NOx and thus this pathway cannot be completely ruled out. Organic nitrate measurements in the southeast US (Lee et al., 2016; Romer et al., 2016) albeit at slightly higher NOx concentrations (although not significantly higher) show that both NO3 and photo-oxidation pathways likely exist. References cited by those papers provide evidence at even lower NOx concentrations. Thus, although there may be reasons that NO3 chemistry may be favored (e.g., higher organic nitrate yields, the correlation with 3-hydroxygluraric acid), photo-oxidation cannot be ruled out and I suggest that this point is presented with a more nuanced view of the complex chemistry that is occurring.

**Response:** Thank you for the comment. The statement is now revised to "Lack of correlation between nitrooxy-organosulfate m/z 294 and 3-hydroxyglutaric acid may indicate a favored nighttime nitrate radical formation pathway over photochemical oxidation. Given that NOx levels at the rural Coweeta sampling site were typically less than 1ppb, photo-oxidation pathways involving high [NOx] to form nitrooxy-organosulfates are less likely. Though a contribution from photochemical oxidation cannot be ruled out (Lee et al., 2016; Romer et al., 2016), nighttime nitrate radical chemistry is most likely the dominating formation mechanism under such conditions."

**Comment**-line 526: The 28% was due to a very specific local biomass burning events, correct? This should be stated here.

**Response**: Yes, the sentence is now revised to "Nitro-aromatics and nitrooxy-organosulfates combined were estimated to account for as much as 28% of WSON for samples impacted by local biomass burning, which reflected the abundance and potential importance of these groups of species to the atmospheric N deposition budget."

**Comment**-line 544: I believe the identified ON/WSON values should be %, not ratio. **Response:** Yes, it is now revised to "the identified ON/WSON percentage".

[revised manuscript text omitted]